# Real-World Data on Clinical Features, Outcomes, and Prognostic Factors in Multiple Myeloma from Miyazaki Prefecture, Japan

**DOI:** 10.3390/jcm10010105

**Published:** 2020-12-30

**Authors:** Keiichi Akizuki, Hitoshi Matsuoka, Takanori Toyama, Ayako Kamiunten, Masaaki Sekine, Kotaro Shide, Takuro Kameda, Noriaki Kawano, Kouichi Maeda, Masanori Takeuchi, Hiroshi Kawano, Seiichi Sato, Junzo Ishizaki, Yuki Tahira, Haruko Shimoda, Tomonori Hidaka, Kiyoshi Yamashita, Yoko Kubuki, Kazuya Shimoda

**Affiliations:** 1Department of Gastroenterology and Hematology, Faculty of Medicine, University of Miyazaki, 5200 Kihara, Kiyotake, Miyazaki 889-1692, Japan; keiichi_akizuki@med.miyazaki-u.ac.jp (K.A.); ayako_kamiunten@med.miyazaki-u.ac.jp (A.K.); masaaki_sekine@med.miyazaki-u.ac.jp (M.S.); koutaro_shide@med.miyazaki-u.ac.jp (K.S.); takuro_kameda@med.miyazaki-u.ac.jp (T.K.); yuuki_tahira@med.miyazaki-u.ac.jp (Y.T.); haruko_shimoda@med.miyazaki-u.ac.jp (H.S.); tmnhdk@med.miyazaki-u.ac.jp (T.H.); yoko_kubuki@med.miyazaki-u.ac.jp (Y.K.); 2Department of Internal Medicine, Koga General Hospital, 1749-1 Sudaki, Ikeuchi Machi, Miyazaki 880-0041, Japan; h-matsuoka@shimane.bc.jrc.or.jp (H.M.); m-takeuchi@kgh.or.jp (M.T.); hiroshi@kgh.or.jp (H.K.); 3Department of Internal Medicine, Miyazaki Prefectural Nobeoka Hospital, 2-1-10 Shinkouji, Nobeoka 882-0835, Japan; t-toyama@pref-hp.nobeoka.miyazaki.jp; 4Department of Internal Medicine, Miyazaki Prefectural Miyazaki Hospital, 5-30 Kitatakamatsu, Miyazaki 880-8510, Japan; kawanoriaki@yahoo.co.jp (N.K.); yamashita@pref-hp.miyazaki.miyazaki.jp (K.Y.); 5Department of Internal Medicine, Miyakonojo Medical Center, 5033-1 Iwayoshi-cho, Miyakonojo 880-8510, Japan; kmtamon@outlook.jp; 6Fujimoto General Hospital, 17-1 Hayasuzumachi, Miyakonojo 885-0055, Japan; mykgmsato@gmail.com; 7Department of Internal Medicine, Miyazaki Aisenkai Nichinan Hospital, 3649-2 Kazeta, Nichinan 887-0034, Japan; jun31_lord@yahoo.co.jp

**Keywords:** real-world outcomes, prognostic factors, multiple myeloma, international staging system

## Abstract

The prognosis of multiple myeloma (MM) has improved with the introduction of novel agents. These data are largely derived from clinical trials and might not reflect real-world patient outcomes accurately. We surveyed real-world data from 284 patients newly diagnosed with MM between 2010 and 2018 in Miyazaki Prefecture. The median follow-up period was 32.8 months. The median age at diagnosis was 71 years, with 68% of patients aged >65 years. The International Staging System (ISS) stage at diagnosis was I in 18.4% of patients, II in 34.1%, and III in 47.5%. Bortezomib-containing regimens were preferred as initial treatment; they were used in 147 patients (51.8%). In total, 80% of patients were treated with one or more novel agents (thalidomide, lenalidomide, or bortezomib). Among 228 patients who were treated with novel agents as an initial treatment, the overall response rate (partial response (PR) or better) to initial treatment was 78.4%, and the median time to next treatment (TTNT) was 11.6 months. In the multivariate analysis, PR or better responses to initial treatment were independently favorable prognostic factors for TTNT. The median survival time after initial therapy for patients with novel agents was 56.4 months and 3-year overall survival (OS) was 70.4%. In multivariate analysis, ISS stage I/II disease and PR or better response to initial treatment, and autologous stem cell transplantation (ASCT) were identified as independent prognostic factors for overall survival (OS).

## 1. Introduction

Multiple myeloma (MM) is a disease characterized by the proliferation of malignant plasma cells. Clinical features include the so-called CRAB criteria (hyperCalcemia, Renal dysfunction, Anemia, Bone lesions) [1]. The prognosis of patients with MM has improved during the past two decades [2]. The introduction of “novel agents” such as thalidomide, lenalidomide, and bortezomib as well as autologous stem cell transplantation (ASCT) has contributed to this improvement, but MM remains incurable with current therapy.

Data demonstrating improvement in MM outcomes are largely derived from clinical trials and might not reflect “real-world” patient outcomes accurately. Several studies based on real-world data have been reported, which are useful for selecting and optimizing treatment in clinical practice [2,3,4,5,6,7,8,9,10]. Studies based on real-world data may be divided into two groups: studies at tertiary referral centers and population-based studies. We conducted a chart review in order to survey the clinical features, therapy options, and outcomes of newly diagnosed MM between 2010 and 2018 in Miyazaki Prefecture, Japan.

## 2. Materials and Methods

### 2.1. Patients and Data Collection

We retrospectively surveyed clinical data of patients newly diagnosed with MM between January 2010 and April 2018 at 7 hospitals in Miyazaki Prefecture, Japan. The ethical approval for this study was obtained from the Research Ethics Committee of the Faculty of Medicine, University of Miyazaki.

### 2.2. Definitions and Clinical Outcome Variables

Clinical staging was based on the International Staging System (ISS) [11]. Standard-risk chromosomal abnormalities (CA) by interphase fluorescence in situ hybridization was defined as absence of del (17p), t (4;14), and t (14;16), whereas high-risk CA was defined as presence of at least one of these abnormalities [12]. Response was based on the International Uniform Response Criteria [13], which includes the following categories: stringent complete response (sCR), complete response (CR), very good partial response (VGPR), partial response (PR), stable response (SD), and progressive disease (PD). Overall response was defined as sCR + CR + VGPR + PR. Overall survival (OS) was calculated from the day of initial therapy to the date of death or last observation. Patients who remained alive at the time of the last follow-up were censored.

### 2.3. Statistical Analysis

Factors that might influence response or survival were analyzed using logistic regression. Categorical variables were first analyzed in univariate analyses. Variables that were significantly associated with response or survival in univariate analyses were included in a Cox proportional hazards model for multivariate analysis. Statistical significance was defined as *p* < 0.05. The Kaplan–Meier method was used to estimate OS. The log-rank test was used to compare OS between groups. All data were analyzed using EZR (Easy R) for the R commander, version 1.37.

## 3. Results

### 3.1. Patients Characteristics at Diagnosis and Initial Treatments

Between January 2010 and April 2018, there were 284 patients diagnosed with MM at 7 hospitals in Miyazaki Prefecture, Japan. Table 1 presents patient and tumor characteristics, initial treatment regimens, and maximum effects. The median age at diagnosis was 71 years (range, 33–93); 68% of patients were aged >65 years. Regarding ISS stage, 47 patients (18.4%) had stage I disease, 87 (34.1%) had stage II disease, and 121 (47.5%) had stage III disease. Data on cytogenetic abnormalities were only available for 93 patients. Among them, 24.6% of patients had abnormalities associated with poor prognosis.

As initial treatment, 37 patients were treated with bortezomib + lenalidomide + dexamethasone, 147 with other bortezomib-containing regimens, 26 with lenalidomide-containing regimens, 18 with thalidomide-containing regimens, and 56 with conventional chemotherapy. The details of conventional chemotherapy are as follows: melphalan + prednisone was used in 37 patients, vincristine + adriamycin + dexamethasone in 6 patients, high-dose dexamethasone in 5 patients, prednisone in 1 patient, and palliative radiation therapy in 7 patients. Overall, 80% of patients were treated with one or more novel agents as bortezomib, lenalidomide, or thalidomide. ASCT was performed in 47 patients (16.5%). The following analysis was focused on 228 patients who were treated with novel agents as an initial treatment. The median follow-up period was 32.8 months.

### 3.2. The Best Response to Initial Treatment

The best response to initial treatment was sCR in 11 patients, CR in 17 patients, VGPR in 49 patients, PR in 68 patients, SD in 29 patients, and PD in 11 patients. Among patients who underwent ASCT, 18.9% achieved CR or better response before ASCT.

The overall response rate, defined as the proportion of patients with PR or better response, was 78.4%. In univariate analysis, age ≤ 65 years (*p* = 0.019) was associated with favorable response.

### 3.3. Time to Next Treatment (TTNT) and 2nd Line Regimens

The median TTNT was 11.6 months for patients treated with novel agents (Figure 1a). TTNT in patients aged ≤65 years was significantly superior to that in patients aged >65 (*p* = 0.011) (Figure 1b). The median TTNT in patients with stage I, II, and III was 10.5, 14.2, and 9.3 months, respectively. TTNT was longer in patients with stage I/II disease compared with those with stage III disease (Figure 1c).

The second line regimens in patients treated with novel agents as an initial treatment were summarized in Table 2. Among patients treated with triplet regimens, 15 (71.4%) patients were secondary treated with next-generation novel agents, such as daratuzumab, elotuzumab, carfilzomib, ixazomib, or pomalidomide. Among patients with bortezomib, 35 (39.3%) and 19 (21.3%) were secondary treated with lenalidomide and next-generation novel agents, respectively. Among patients with lenalidomide, 5 (50.0%) were secondary treated with bortezomib.

### 3.4. Overall Survival

The median survival time (MST) since initial treatment was 56.3 months (Figure 2a). The MST for patients aged ≤65 years was 96.3 months, and that for patients aged >65 years was 48.2 months (*p* < 0.001) (Figure 2b). MST for patients with stage I, II, and III disease was 110.0, 72.0, and 42.2 months, respectively (*p* < 0.001) (Figure 2c). Of 228 patients, 44 patients underwent ASCT. Patients who underwent ASCT had longer MST (72.0 months) than patients who did not undergo ASCT (40.4 months) (*p* < 0.001) (Figure 2d). In the 84 patients aged ≤65 years, 34 patients underwent ASCT while the remaining 50 did not. Reasons for not undergoing ASCT were available for 41 patients. The most frequent reasons were accessibility difficulties to the transplant facility (*n* = 14, 34.1%), followed by impaired major organ functions (*n* = 9, 22.0%: renal dysfunction in 5, respiratory dysfunction in 1, cardiac dysfunction in 3), no patient wish to undergo transplantation (*n* = 7, 17.1%), poor disease control (*n* = 5, 12.2%), lower performance status (*n* = 3, 7.3%), and others (complication of solid tumors, poor mobilization, intractable diverticulitis; each 1 case).

### 3.5. Factors Which Affected on TTNT and OS

Among patients who were treated with novel agents as an initial treatment, both TTNT and OS were significantly better in the deeper response groups by response to initial treatment. TTNT in patients achieving PR or better response was 15.3 months, whereas TTNT of patients achieving SD or PD was 6.6 months (*p* < 0.001) (Figure 3a). MST in patients achieving PR or better response was 68.2 months, whereas MST of patients achieving SD or PD was 39.6 months. Patients achieving PR or better response after the initial treatment had better survival than patients with SD or PD (*p* < 0.001) (Figure 3b).

We analyzed the effect of initial therapy response and ASCT together with patient characteristics. In univariate analyses, age ≤ 65 year, ISS stage I/II stage, PR or better response to initial treatment, and ASCT were associated with longer TTNT, and age ≤65 years, ISS stage I/II disease, PR or better response to initial treatment, and ASCT were associated with favorable OS (Table 3 and Table 4). In multivariate analysis, PR or better responses to initial treatment were independently favorable prognostic factors for TTNT, and ISS stage I/II disease and PR or better response to initial treatment, and ASCT were independent favorable prognostic factors for OS.

## 4. Discussion

We retrospectively surveyed patient characteristics, clinical features, treatment options, and outcomes in patients newly diagnosed with MM. These data provide valuable information on the circumstances of MM and real-world outcomes following treatment and factors that influence prognosis.

The median age at diagnosis of our patients was 71 years, and 68% of patients in this study were aged >65 years. These results were similar to those in previous real-world reports from a European multicenter observational study and a Japanese regional study [3,4], but the median age of our patients was approximately 5 years older than in previous reports from the Mayo Clinic, a Japanese Society of Myeloma (JSM) multicenter study, and Czech registry data [2,5,6]. In our cohort, only 18.4% of patients had ISS stage I disease at diagnosis. This proportion was similar to the proportion in the European observational study (16%), but was much lower than that in reports from the Mayo Clinic (30%), JSM multicenter study (26.5%), and Czech registry data (31.9%) [2,4,5,6]. Older age was correlated with higher ISS stage, with 10.9% of patients aged >65 years presenting with ISS stage I disease in our cohort. This was also observed in the Czech registry data, where only 12% of patients aged ≥80 years presented with ISS stage I disease [5]. The lower proportion of patients with ISS stage I disease in both our cohort and the European observational study might be due to the higher proportion of elderly patients compared with other reports.

Current guidelines recommend initial treatment with one or more novel agents and ASCT for eligible patients. In our cohort, about 80% of patients received one or more novel agents as the initial treatment. In contrast, only 16.5% of patients underwent ASCT, which was much lower than that in the Mayo Clinic reports or JSM multicenter study, in which 37% and 34% of patients received ASCT, respectively [2,6]. One reason for this difference might be the larger proportion of elderly patients in our cohort. In fact, this gap narrowed when the analysis was restricted to patients aged ≤65 years; 40.0% and 56% of patients aged ≤65 years received ASCT in our cohort and the Mayo Clinic cohort, respectively, although the gap was still present. Another major reason was the difficulty in accessing transplant facilities.

Among patients who were treated with novel agents as an initial therapy, most (78.4%) achieved PR or better response after initial treatment. The proportion of patients who achieved CR (9.2%) and VGPR or better response (41.6%) in our cohort was much lower than that in the European observational study, in which 32% and 74% of patients achieved CR and VGPR or better response, respectively [4]. In addition, the TTNT in our cohort was 11.6 months, which was also inferior to the corresponding value (progression-free survival: 16.3 months) in the European observational study. We are not able to account for this difference of a deeper response (CR/VGPR) and TTNT between our cohort and the European observational study at this time. However, more elderly patients in our cohort might account for the difference. Even though the response rate in our cohort was lower and TTNT was shorter than those reported in other real-world studies, OS seemed to be similar. MST in our cohort was 56.4 months, which was comparable with MST of 50.3 months in the Czech registry data and 51.8 months in a previous Japanese regional study [3,5], but was inferior to the 6.1 years or 60.6 months in reports from the Mayo Clinic or JSM multicenter study [2,6]. In univariate analysis, age ≤ 65 years, ISS stage I/II disease, PR or better response to initial treatment, and ASCT were associated with favorable prognosis. The inferior OS in our cohort compared to reports from the Mayo Clinic or JSM multicenter study might be due to the higher proportion of elderly patients, lower proportion of patients with ISS stage I disease, and lower proportion of patients who underwent ASCT. Patient characteristics in our cohort and outcomes resembled the results of the European observational study and Czech registry data, which might reflect real-world practice for MM.

In addition to patient age, disease burden, and treatment choice, which are all based on pre-treatment status, PR or better response to initial treatment was associated with longer TTNT and better OS, which was consistent with many previous reports [6,14]. This initial treatment effect was also observed with multivariate analysis.

This study had some limitations. It was retrospective in design. Each physician determined the choice of treatment, and treatment doses and durations varied across individuals. Physician definitions of treatment response may not be as rigorous as the evaluation criteria used in clinical trials. However, our data constituted a population-based cohort. Therefore, the results of this study reflect daily practice in the era of novel agents. Finally, early identification of patients with poor prognosis after the initial treatment will allow us to change the therapeutic strategy earlier, which may contribute to better survival.

## Figures and Tables

**Figure 1 jcm-10-00105-f001:**
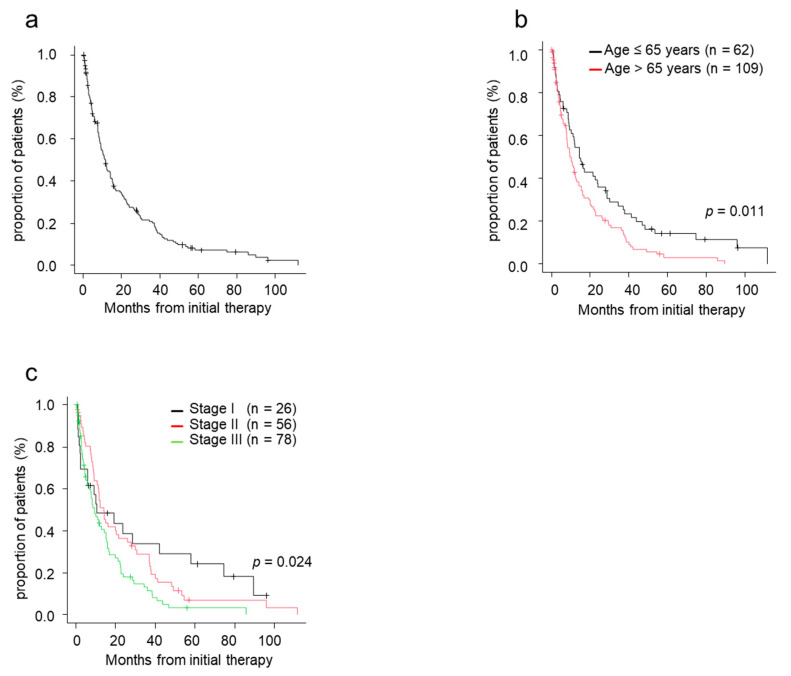
Time to next treatment (TTNT) by patient characteristics. (**a**) TTNT from initial therapy. (**b**) TTNT from initial therapy by age. (**c**) TTNT from initial therapy by ISS stage. ISS, International Staging System; TTNT, time to next treatment.

**Figure 2 jcm-10-00105-f002:**
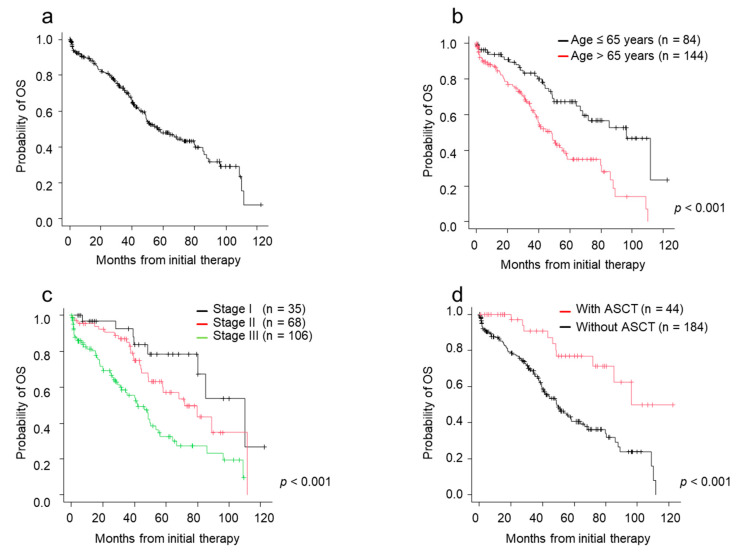
Overall survival (OS) by patient characteristics and type of therapies. (**a**) OS from initial therapy. (**b**) OS from initial therapy by age. (**c**) OS from initial therapy by ISS stage. (**d**) OS from initial therapy by ASCT status. ASCT, autologous stem cell transplantation; ISS, International Staging System; OS, overall survival.

**Figure 3 jcm-10-00105-f003:**
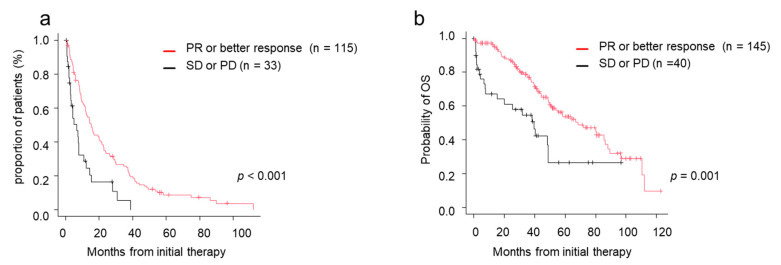
Time to next treatment (TTNT) and overall survival (OS) by response to initial therapy. (**a**) TTNT in those who achieved PR/better versus those who did not. (**b**) OS in those who achieved PR/better versus those who did not. CR, complete response; OS, overall survival; PD, progressive disease; PR, partial response; sCR, stringent complete response; SD, stable disease; TTNT, time to next treatment; VGPR, very good partial response.

**Table 1 jcm-10-00105-t001:** Patient characteristics and clinical summary.

Characteristic	Overall	Non-Novel Agent		Novel Agent		*p* Value ^#^	*p* Value ^☥^
			Total	Age ≤ 65 years	Age > 65 years		
	(*n* = 284)	(*n* = 56)	(*n* = 228)	(*n* = 84)	(*n* = 144)		
Age, y	71.0 (33–93)	75 (42–90)	69 (33–93)	58.5 (33–65)	77 (66–93)	0.002	<0.001
Sex							
Male	144 (50.7)	25 (44.6)	119 (52.2)	45 (53.6)	74 (51.4)	0.388	0.857
Female	140 (49.3)	31 (55.4)	109 (47.8)	39 (46.4)	70 (48.6)		
ISS stage						0.073	0.002
I	47 (18.4)	12 (26.1)	35 (16.7)	21 (26.2)	14 (10.9)		
II	87 (34.1)	19 (41.3)	68 (32.5)	29 (36.2)	39 (30.2)		
III	121 (47.5)	15 (32.6)	106 (50.7)	30 (37.5)	76 (58.9)		
Missing data	29	10	19	4	15		
Cytogenetic profile							
Standard-risk ^a^	70 (75.4)	4 (66.7)	67 (77.0)	24 (80.0)	43 (75.4)	0.987	0.687
High-risk ^b^	23 (24.6)	2 (33.3)	20 (23.0)	6 (20.0)	14 (24.6)		
Missing data	191	50	141	54	87		
Initial therapy						<0.001	<0.001
Triplet	37 (13.0)	0 (0.0)	37 (16.2)	12 (14.3)	25 (17.4)		
Bortezomib-containing regimen	147 (51.8)	0 (0.0)	147 (64.5)	69 (82.1)	78(54.2)		
Lenalidomide-containing regimen	26 (9.6)	0 (0.0)	26 (11.4)	3 (3.6)	23 (16.0)		
Thalidomide-containing regimen	18 (6.3)	0 (0.0)	18 (7.9)	0 (0.0)	18 (12.5)		
MP regimen	37 (13.0)	37 (66.1)	0 (0.0)	0 (0.0)	0 (0.0)		
Other	19 (6.7)	19 (33.9)	0 (0.0)	0 (0.0)	0 (0.0)		
ASCT						0.021	<0.001
Yes	47 (16.5)	3 (5.4)	44 (19.3)	34 (40.5)	10 (6.9)		
No	237 (83.5)	53 (94.6)	184 (80.7)	50 (59.5)	134 (93.1)		
Best response						0.004	<0.001
OR (sCR + CR + VGPR + PR)	164 (74.9)	19 (55.9)	145 (78.4)	59 (88.1)	86 (72.9)		
sCR	11 (5.0)	0 (0.0)	11 (5.9)	4 (6.0)	7 (5.9)		
CR	17 (7.8)	0 (0.0)	17 (9.2)	10 (14.9)	7 (5.9)		
VGPR	56 (25.5)	7 (20.6)	49 (26.5)	14 (20.9)	35 (29.7)		
PR	80 (26.2)	12 (35.3)	68 (36.8)	31 (46.3)	37 (31.4)		
SD	41 (18.7)	12 (35.3)	29 (15.7)	6 (9.0)	23 (19.5)		
PD	14 (6.4)	3 (8.8)	11 (5.9)	2 (3.0)	9 (7.6)		
Missing data	65	22	43	17	26		

^a^, Standard risk defined as absence of high risk chromosomal abnormalities; ^b^, High risk chromosomal abnormalities defined as presence of del 17p and/or t (4;14) and/or t (14;16); ASCT, autologous stem cell transplantation; CR, complete response; ISS, International Staging System; MP, melphalan and prednisone; NA, not available; OR, overall response; PD, progressive disease; PR, partial response; sCR, stringent complete response; SD, stable disease; VGPR, very good partial response. *p*^#^, *p* value between patient treated with novel agent and patients treated without novel agent; *p*^☥^, *p* value between patient aged ≤65 years and patients aged >65 years.

**Table 2 jcm-10-00105-t002:** Second line regimens in patients treated with novel agents as an initial treatment.

Second Line Treatment Regimen	Initial Treatment Regimen			
	Triplet (*n* = 37)	Bortezomib-Containing (*n* = 147)	Lenalidomide-Containing (*n* = 26)	Thalidomide-Containing (*n* = 18)
Daratuzumab-containing	3 (14.3)	0 (0.0)	0 (0.0)	0 (0.0)
Elotuzumab-containing	4 (19.0)	3 (3.4)	0 (0.0)	0 (0.0)
Carfilzomib-containing	3 (14.3)	2 (2.2)	1 (10.0)	0 (0.0)
Ixazomib-containing	1 (4.8)	3 (3.4)	1 (10.0)	0 (0.0)
Bortezomib-containing	1 (4.8)	16 (18.0)	5 (50.0)	3 (30.0)
Pomalidomide-containing	4 (19.0)	11 (12.4)	1 (10.0)	0 (0.0)
Lenalidomide-containing	3 (14.3)	35 (39.3)	0 (0.0)	6 (60.0)
Thalidomide-containing	0 (0.0)	1 (1.1)	0 (0.0)	1 (10.0)
Triplet	2 (9.5)	12 (13.5)	2 (20.0)	0 (0.0)
Conventional chemotherapy	0 (0.0)	6 (6.7)	0 (0.0)	0 (0.0)
No further treatment	5	12	3	2
due to poor general conditions *	0	3	2	2
due to unknown reasons	5	9	1	0
Early death	3	9	5	1
Missing data	8	37	8	5

* Poor general conditions including low performance status, infectious complications, or organ dysfunctions resulting from 1st line treatment.

**Table 3 jcm-10-00105-t003:** Univariate and multivariate analysis of clinical factors influencing time to next treatment from initial therapy.

Variable	Univariate		Multivariate	
	HR (95% CI)	*p* Value	HR (95% CI)	*p* Value
Age (>65 vs. ≤65 years)	1.55 (1.10–2.18)	0.011	1.22 (0.79–1.89)	0.367
Stage (III vs. I or II)	1.58 (1.12–2.26)	0.005	1.37 (0.94–1.99)	0.090
Cytogenetic profiles (High-risk ^a^ vs. standard-risk ^b^)	1.22 (0.66–2.28)	0.523		
ASCT status (performed vs. not performed)	0.64 (0.42–0.98)	0.041	0.88 (0.52–1.48)	0.630
Response of initial therapy (PR or better vs. SD or PD)	0.43 (0.27–0.66)	<0.001	0.50 (0.31–0.79)	0.003

Variables significant in univariate models were included in multivariate Cox regression model. ^a^, High-risk chromosomal abnormalities defined as presence of del (17p) and/or t (4;14) and/or t (14;16); ^b^, Standard-risk defined as absence of high risk chromosomal abnormalities; ASCT, autologous stem cell transplantation; CI, confidence interval; HR, hazard ratio; PD, progressive disease; PR, partial response; SD, stable disease.

**Table 4 jcm-10-00105-t004:** Univariate and multivariate analysis of clinical factors influencing overall survival from initial therapy.

Variable	Univariate		Multivariate	
	HR (95% CI)	*p* Value	HR (95% CI)	*p* Value
Age (>65 vs. ≤65 years)	2.45 (1.56–3.84)	<0.001	1.30 (0.76–2.22)	0.332
Stage (III vs. I or II)	2.57 (1.66–4.00)	<0.001	2.19 (1.37–3.49)	0.001
Cytogenetic profiles (High-risk ^a^ vs. standard-risk ^b^)	1.57 (0.74–3.31)	0.239		
ASCT status (performed vs. not performed)	0.30 (0.16–0.58)	<0.001	0.40 (0.18–0.85)	0.017
Response of initial therapy (PR or better vs. SD or PD)	0.43 (0.26–0.71)	0.001	0.47 (0.27–0.79)	0.005

Variables significant in univariate models were included in multivariate Cox regression model. ^a^, High-risk chromosomal abnormalities defined as presence of del (17p) and/or t (4;14) and/or t (14;16); ^b^, Standard-risk defined as absence of high risk chromosomal abnormalities; ASCT, autologous stem cell transplantation; CI, confidence interval; HR, hazard ratio; PD, progressive disease; PR, partial response; SD, stable disease.

## Data Availability

The data that support the findings of this study are available on request from the corresponding author. The data are not publicly available due to privacy or ethical restrictions.

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
