# Peer review of "Real-World Data on Clinical Features, Outcomes, and Prognostic Factors in Multiple Myeloma from Miyazaki Prefecture, Japan"

_jcm, 2020, doi:10.3390/jcm10010105_

Round 1
Reviewer 1 Report
This is an interesting paper of real world data from Japan. This alone warrants consideration given the lack of non-North American or European data in this space. As the authors point out, this cohort is enriched for non-ASCT treated patients and this factor is an important consideration in interpreting the data. The median OS is largely as expected given the spectra of therapies used. Overall, it is an important snapshot of current outcomes in Japan and complements existing data for the country.
A few points would strengthen the manuscript:
- If data was available it would be important to clarify/characterize the next line(s) of therapy that all influence OS. This is particularly important given (as the authors point out) the depth of responses based on frontline therapy are lower than expected. Even just being able to add in either PFS or time to next therapy (TTNT) would help.
- Needs further discussion about the lack of ASCT in what seem to be otherwise eligible patients (i.e. < 65y… arguably < 70y in many places). Interestingly the median OS for the ASCT cohort (although small) is what one would expect in a modern cohort. On that note it would be important to state whether the transplant cohort received maintenance therapy.
- It would be good to provide a survival curve for NA vs non-NA (especially for the >65y). The non-NA cohort is largely historical and has little relevance for modern MM therapy other than to provide context for an extended cohort such as this. In fact, to make this relevant to current practice one might consider re-analysing the data with the non-NA patients excluded.
- I am not sure the data summarized in figure 3 adds much. The inclusion of SD as prognostic and of clinical relevance that might inform practice does not clinically have much bearing on baseline ISS stage. This feature alone often prompts a change in therapy. I would probably not include in the final manuscript in its present form. That said, examining outcome as it might pertain to a change in therapy would be key. Again, the data on subsequent therapy is important in examining this.
- The authors must clearly describe how response depth was assigned. Is this calculated by the authors based on source laboratory data, lab values described by the treating physician in the health record or an inferred response rate as per the treating physicians description in the health record (such as the clinical note).
Minor issues:
- Figure 2 alludes to treatment abbreviations but no treatment-related data is discussed
- The authors should review for typographical errors (ex. Line 141 us of “of on”, line 149 favourite should be favourable)
I would recommend (space permitting) to ask the authors to incorporate more data on PFS, TTNT and subsequent therapies ( at least line 2), a more focused examination of the NA-treated patients (perhaps even only the NA treated patients to increase modern clinical relevance) and further discussion on why ASCT was not pursued more frequently. This ultimately make the paper more useful in guiding practice.
Reviewer 2 Report
The authors performed a retrospective analysis of MM patients diagnosed and treated in Japan in a 8-year span between 2010 and 2018. They reported baseline characteristics as well as first line therapy details and eventually survival outcomes.
The manuscript is well written but this report has some major flows and does not add a significant contribute to the available literature; in the current treatment landscape of NDMM patients, a small proportion of patients received ASCT, 1/5 of older patients only received a palliative therapy rather than active chemotherapy; therefore this population does not seem representative of the current clinical scenario.
Many of the reported data are well established information and do not improve the current knowledge (e.g. the correlation between ISS and response to treatment to OS).
COMMENTS:
- despite an observation span of 8 years the median follow-up is only 18 months; can the authors comment on that?
- I would add the following info to the baseline characteristics when possible: MDE that drove treatment initiation; RISS; 2nd line and following treatment lines
- which cut-off were used to define positivity for 4;14, 14;16, del17p? the % of HR patients is higher than the expected (30% vs 15%)
- it is striking that 60% of patients despite being younger than 65 years did not receive ASCT as per standard of care; can the authors comment on that? also, 1/5 of older patients only received palliative therapies.
- the authors reported in the univariate analysis that HR FISH analysis did not have a significant impact on OS; this is hard to believe as HR CA is probably the most powerful prognostic factor. Can they explain that?
Reviewer 3 Report
Good presentation of clinically relevant characteristics and outcomes data of patients with MM from real world experience. Good write-up and nice discussion over all with minor suggestions:
- You said that among 90 patients younger than 65 years, 36 pts got auto-SCT and 55 did not. The sum doesn't add up to 90. Please correct that.
- In lines 179-181, think you missed to mention ISS stage "1". Please include that.
- Could the lower response rates to treatment in your study group compared to the European group be due to the lower percentage of use of novel agents in your group? Please shed light on that.
- Any available data on the causes for the lower percentage of patients younger than 65 years getting stem cell transplant in your group? Poor access or availability of stem cell transplant? distance from facilities offering stem cell transplant? low socioeconomic status? lack of health care funding? Any info on these factors will enhance the quality of the study as these factors have been shown to determine the SCT in patients with MM and also the outcomes.
- In the lines 210-212, saying that early change in therapeutic strategy is warranted for patients with advanced stage disease and poor initial treatment response is too strong and without strong evidence from a study with retrospective design. Also in the lines 217-219, saying prospective clinical trials testing the strategy of changing treatment based on lack of significant initial response [PR or better] to improve outcomes is scientifically more appropriate than just recommending change of treatment based on the findings from a retrospective study.
- Given the p-value being close to 0.05 [0.056], you could consider saying that SCT showed a trend towards improved survival in multivariate analysis [?]
Round 2
Reviewer 2 Report
The authors nicely addressed all the issues raised thus making the manuscript more informative for readers.
One last suggestion is to cut the 2nd line section with all the 2nd line treatments listed as it is hard to read; a table would be more appropriate.
